# Geometric frustration in polygons of polariton condensates creating vortices of varying topological charge

Tamsin Cookson [1,2], Kirill Kalinin[1,3], Helgi Sigurdsson [1,2], Julian D. Töpfer[1,2], Sergey Alyatkin[1], Matteo Silva[2], Wolfgang Langbein [4], Natalia G. Berloff[1,3 ✉] & Pavlos G. Lagoudakis [1,2 ✉]

Vorticity is a key ingredient to a broad variety of fluid phenomena, and its quantised version is considered to be the hallmark of superfluidity. Circulating flows that correspond to vortices of a large topological charge, termed giant vortices, are notoriously difficult to realise and even when externally imprinted, they are unstable, breaking into many vortices of a single charge. In spite of many theoretical proposals on the formation and stabilisation of giant vortices in ultra-cold atomic Bose-Einstein condensates and other superfluid systems, their experimental realisation remains elusive. Polariton condensates stand out from other superfluid systems due to their particularly strong interparticle interactions combined with their non-equilibrium nature, and as such provide an alternative testbed for the study of vortices. Here, we non-resonantly excite an odd number of polariton condensates at the vertices of a regular polygon and we observe the formation of a stable discrete vortex state with a large topological charge as a consequence of antibonding frustration between nearest neighbouring condensates.

[1] Skolkovo Institute of Science and Technology, Skolkovo, Russian Federation. [2] Department of Physics and Astronomy, University of Southampton, Southampton, UK. [3] Department of Applied Mathematics and Theoretical Physics, University of Cambridge, Cambridge, UK. [4] School of Physics and Astronomy, Cardiff University, Cardiff, UK. ✉email: N.G.Berloff@damtp.cam.ac.uk; P.Lagoudakis@skoltech.ru

Quantised vortices are fundamental topological objects playing an important role in branches of physics ranging from superfluids and superconductors to high-energy physics and optics. They exist in classical matter fields, described by a complex-valued function $\Psi(\mathbf{r}, t) = \sqrt{\rho(\mathbf{r}, t)} \exp[iS(\mathbf{r}, t)]$, as singular points in two dimensions, or lines in three dimensions, where $\rho = 0$ and the phase $S$ winds around in multiples of $2\pi$. The winding of a quantised vortex, also called topological charge, is the integer defined as $m = (1/2\pi) \oint_C \nabla S \cdot d\mathbf{l}$, where $C$ is the closed contour around the vortex singularity. Although the formation, structure, dynamics, and turbulence of quantised vortices have been the subject of intense research[1], many fundamental aspects of vortex dynamics such as its effective mass[2] and applicable forces[3] are still not fully understood. Vortex motions, even in the simplest configurations, such as the advection of a single vortex of unit charge by a constant superflow have intrigued the scientific community[4].

The creation of multiply quantised vortices, or giant vortices, in ultra-cold atomic Bose–Einstein condensates (BECs) has challenged researchers[5–8] since a multiply charged vortex, both in a uniform and sufficiently large condensate[9], as well as in a trapped condensate[10], is dynamically and thermodynamically unstable and tends to break into an array of singly charged vortices[11]. Numerous theoretical proposals address how to stabilise multiply charged vortices, such as exploiting stronger than harmonic confinement and fast rotation[6,7,12], using a near-resonant laser beam[8], introducing a second species of BEC[13], organising spatiotemporally modulated interactions[14], or with spin–orbit coupling[15]. However, evidence that many vortices can merge and stay stable as a single giant vortex remains elusive, as is the case in other quantum fluid systems such as superfluid helium-3 and 4[16,17].

In mesoscopic superconducting materials, vortices of large topological charge have already been reported[18–20] with numerical simulations corroborating the evidence, since the available experimental techniques provided limited probing of the vortex core[20–22]. In many respects, these vortices have similar physics to giant vortices in atomic BECs but with external rotation controlled through an applied magnetic field. By increasing the amplitude of the magnetic field, a vortex of multiplicity three was experimentally observed in a strongly confined type-II superconductor[23]. Apart from quantised vortices that exist on a non-zero, and usually uniform background, a new class of vortices was introduced theoretically and achieved experimentally in optics and atomic physics: those found in optical lattices of ultra cold BECs or periodic photonic structures of light called discrete vortex solitons (DVSs)[24–26]. The core of such vortices lies on a negligible density background and their phase winds to provide spatially localised circular energy flows between lattice sites. In previous experimental realisations, the phase corresponding to multiple singly charged vortices was directly imprinted by laser beams that resulted in stationary DVSs[27], and a double-charged vortex was shown in a hexagonal photonic lattice[28]. Similarly, a ring network of dissipatively coupled lasers[29,30] or optical parametric oscillators[31] may establish a phase winding as an excited state. Non-equilibrium condensates, on the other hand, offer alternative mechanisms for stabilisation of giant vortices such as inward particle fluxes towards the trapped condensate's centre[32].

Geometric frustration (henceforth frustration) occurs in magnetic systems when the relative arrangement of spins possess multiple degenerate ground-state configurations[33]. It has been intensely studied since 1950, when its formation was presented in a two-dimensional (2D) Ising net in a triangular lattice[34] and later in the three-dimensional pychlore lattice[35]. Frustration can

be engineered into systems by controlling the geometry and coupling strength between spins. Atoms trapped in a 2D optical lattice were able to simulate different regimes of the $XY$ Hamiltonian[36]. Similarly, frustration has also been seen in lattices of coupled laser systems. A negatively coupled triangular lattice with $\pm 2\pi/3$ phase locking has been observed, leading to singly quantised vortices and alternate chirality in the lattice[37]. In strongly coupled microcavity polariton lattices, frustration caused the observation of a flat band in the dispersion of a Lieb lattice[38].

In this article, we realise frustrated polygons of polariton condensates as a platform to create and study circulating currents leading to vortices of varying topological charge. Polaritons are solid-state quasiparticles that result from the hybridisation of strongly coupled excitons and photons in semiconductor microcavities. At low densities, they behave as short-lived bosons (several picoseconds lifetime) that can undergo a power-driven phase transition into a non-equilibrium condensate[39] remaining indefinitely stable under continuous non-resonant excitation. Polariton condensates and related phenomena have been the subject of intense investigation for some time[40,41], including skyrmions[42] and quantised vortices and their dynamics[43–52]. Unlike DVSs in purely optical systems or photonic crystals, where the phase winding comes from the laser, in polariton graphs, vortices form spontaneously during condensation. In this respect, they are closer to vortices in atomic condensates, but as we demonstrate, they do not require any external stirring, and even multiply charged vortices are fully stabilised in the non-equilibrium polariton system. Also, unlike vortices in ultra-cold atomic BECs, polariton giant vortices generate spiral velocity profiles towards the vortex centre[53]. Here, we consider ballistically expanding condensates at the vertices of a regular polygon. Figure 1a shows a schematic of the microcavity system with a pentagon non-resonant pumping geometry (red cones). The term "ballistically expanding" means that the condensates are not confined in a potential minimum but are instead gain-localised due to the tightly focussed (~2 μm full width at half maximum) optical excitation beams feeding particles into the condensates. The advantage of such non-Hermitian localisation is the coherent polariton outflow from each pump spot, causing strong interference with neighbouring condensates[54].

## Results

By incrementally raising the polygon's pump power, polaritons eventually undergo condensation and the system is described by an order parameter $\Psi(\mathbf{r}, t)$ belonging to a polariton mode with the highest optical gain. By tuning the separation distance between pump spots, one can control the coupling between nearest neighbours[55], and therefore, the phase configuration across the polygon. Although we present results on a single macroscopic coherent polariton condensate across the entire polygon, we will refer also to the vertices of the polygon as "individual" condensates, which can interfere and synchronise. The modes of the entire system belong to a point symmetry operator $\mathcal{C}_N$ of cyclic $N$-fold rotations, where $N$ is the number of condensates (vertices in the polygon). Being rotationally periodic, one can apply Bloch's theorem along the planar azimuthal angle $\varphi$ such that $\Psi_{q,n}(\mathbf{r}) = e^{iq\varphi} u_{q,n}(\mathbf{r})$, where $u_{q,n}(r, \varphi) = u_{q,n}(r, \varphi + 2\pi/N)$ is the Bloch mode of the system and $q = 2\pi m/L$ is the quasimomentum along the polygon's circumcircle for $m \in \mathbb{Z}$ satisfying $|m| \le N/2$ and $L = Na$ being the length of the polygon's circumcircle. In Fig. 1b, c, we show example of two energy branches of the discrete Bloch modes (red dots) for $N = 5$ and $N = 6$ polygon, respectively, in the reduced Brillouin zone. The blue curves denote the bands in the thermodynamic limit $N \to \infty$.

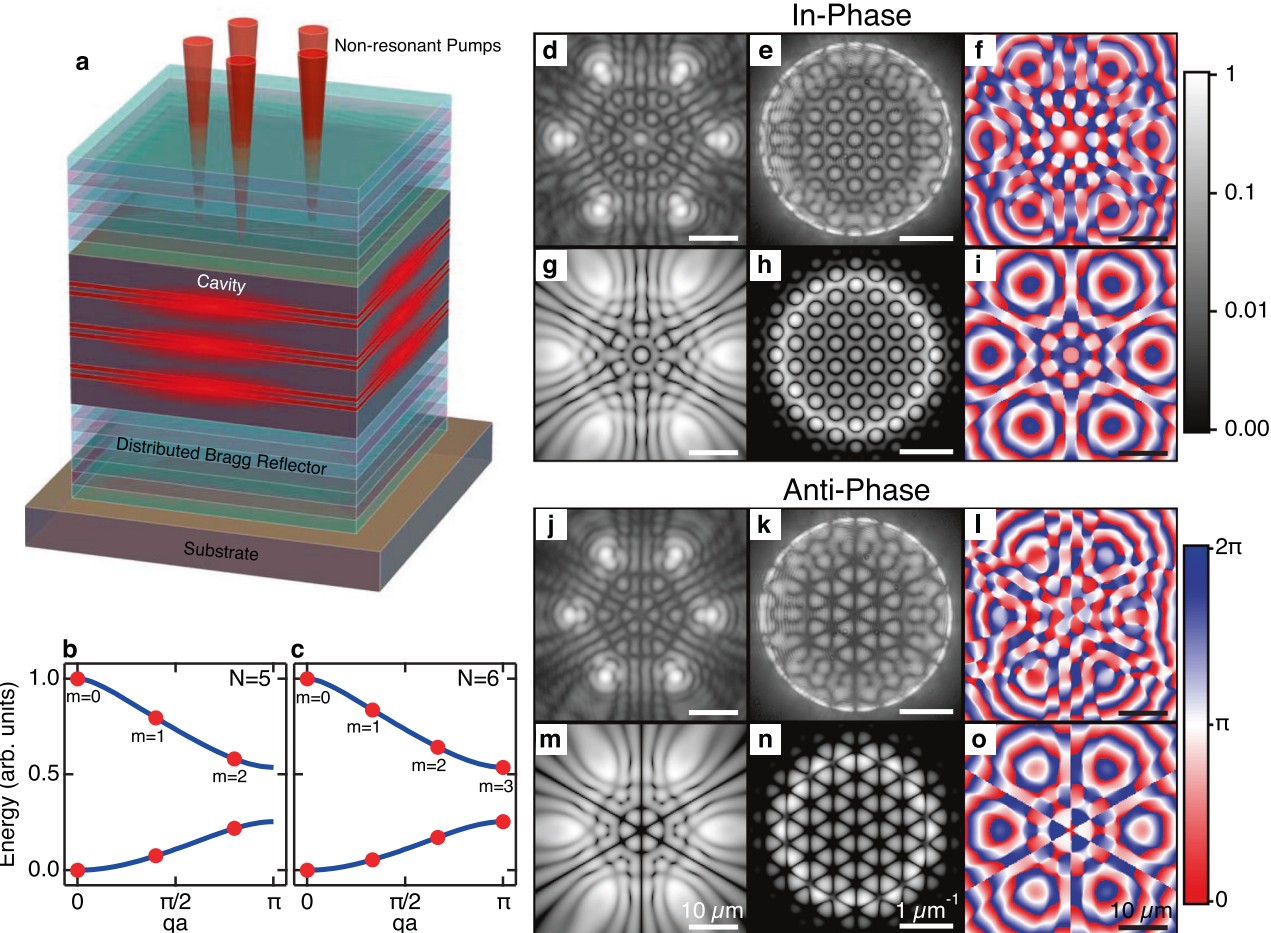

**Fig. 1 Excitation schematic, Bloch modes, and stationary states in hexagons of polariton condensates. a** Schematic showing a quantum well microcavity. Non-resonant lasers (red cones) at normal incidence create polariton condensates at their respective locations. **b**, c Example of reduced Brillouin zone energies of angular Bloch modes (red dots) in an $N = 5$ and $N = 6$ polygon, respectively. Blue curves show band structure in the thermodynamic limit. **d**, **j** Experimental condensate photoluminescence in real-space and **e**, **k** Fourier space in a hexagon geometry, with radii $R = 16.9\ \mu m$ and $R = 14.7\ \mu m$, showing in-phase and anti-phase locked condensates, respectively. **f**, **l** Corresponding experimentally extracted real-space polariton phase maps. **g–i** and **m–o** show simulations of the steady-state polariton condensate wavefunction $\Psi(\mathbf{r}, t)$ using the driven-dissipative Gross–Pitaevskii equation in the in-phase and anti-phase locked configuration, respectively. **g**, **m** Real-space density $|\Psi(\mathbf{r})|^2$, **h**, **n** Fourier-space density $|\hat{\Psi}(\mathbf{k})|^2$, and **i**, **o** phase $\arg(\Psi(\mathbf{r}))$. All real-space, Fourier-space, and phase images are plotted on the same scale defined on the scale bar at the bottom of each column.

Previously, it was shown that condensation of ballistically expanding condensates can be correlated with the ground-state configuration of the *XY* Hamiltonian, $\mathcal{H}_{XY} = -\sum_{ij} J_{ij} \cos(\theta_i - \theta_j)$, where $J_{ij}$ and $\theta_{ij} = \theta_i - \theta_j$ are the coupling strength and the relative phase between condensates $i$ and $j$, respectively[55–57]. The coupling strength $J_{ij}$ depends on the pump power, the distance between condensates, $d_{ij} = |\mathbf{r}_i - \mathbf{r}_j|$, and the in-plane wavenumber $k_c$ of the condensate polaritons expanding from their respective pump spots[54], all of which can be controlled by tuning the pumping power and geometry using spatial light modulators (SLMs). If one tunes the system to have $J_{ij} > 0$, then the coupling is said to be ferromagnetic and a solution with maximum polariton gain has all condensates locked in-phase, whereas if $J_{ij} < 0$, the coupling is said to be antiferromagnetic and the condensates try to be in anti-phase ($\pi$ phase). In a regular polygon with uniform nearest neighbour interactions, the *XY* Hamiltonian becomes cyclic $\mathcal{H}_{XY} = -J \sum_{i=1}^{N} \cos \theta_{i,i+1}$, with the boundary condition $\theta_1 = \theta_{N+1}$. It is then understood that in-phase and anti-phase configurations are simply modes with quasimomentum $q = 0$ and $q = N\pi/L$, at the centre and edges of the crystal Brillouin zone, respectively,

where polaritons have been observed to preferentially condense into[58]. Correspondingly, Bloch modes between the centre and the edges of the Brillouin zone, i.e. $0 < |m| < N/2$, are simply vortices of winding number $m$.

When $J < 0$, and $N$ is odd, the edges of the Brillouin zone become forbidden and polaritons condense instead into the state closest to the edge, which maximises the gain. For odd numbered $N$, this state is $\theta_{i,i+1} = \pm (N-1)\pi/N$. Interestingly, a superposition of the two counter-propagating modes is not observed as stable solution of the frustrated polygon system, but instead, a breaking of the parity symmetry occurs with the formation of a net polariton current along the polygon circumcircle. Such symmetry breaking can only be attributed to nonlinear effects in the condensate and has previously been reported in both experiment[59,60] and theory[61–63]. Solutions of lower vorticity (further away from the Brillouin zone edge) are also stable and observable in measurements even though they do not correspond to the highest gain of the *XY* model. Here, we report on the observation of these parity-breaking solutions forming discrete vortex (or simply *vortex*) states in polygon configurations of polariton condensates with winding number $m$. In the simplest

case of $N = 3$, the formation of a $|m| = 1$ vortex ($\theta_{i,i+1} = \pm 2\pi/3$) was observed in 2016[55].

To experimentally control the condensate phase configurations, we inject equidistant polariton condensates at the vertices of a regular polygon. We preclude any correlation between the phase of the pumping source and the realised phase configurations, by pumping polariton condensates using non-resonant continuous wave (CW) optical excitation on a multiple InGaAs quantum well (QW) semiconductor microcavity[64] (see Fig. 1a). For a description of the sample, and excitation and detection schemes, see "Experimental techniques" in "Methods". In addition, we simulate the dynamics of the polariton condensates using the 2D driven-dissipative Gross–Pitaevskii equation[65,66] written for the condensate wavefunction, $\Psi(\mathbf{r}, t)$, and the rate equation of the hot exciton reservoir, $X(\mathbf{r}, t)$:

$$i\hbar\frac{\partial \Psi}{\partial t} = \left[(i\Lambda - 1)\frac{\hbar^2}{2m^*}\nabla^2 + \alpha|\Psi|^2 + gX + GP(\mathbf{r}) + \frac{i\hbar}{2}(RX - \gamma)\right], \quad (1)$$

$$\frac{\partial X}{\partial t} = -(\Gamma + R|\Psi|^2)X + P(\mathbf{r}), \quad (2)$$

where $m^*$ is the polariton effective mass, $\alpha$ is the polariton–polariton interaction strength, $g$ is the interaction strength of polaritons with the exciton reservoir feeding the condensate, $G$ is the interaction strength of polaritons with the dark background of inactive excitons that do not scatter into the condensate, $R$ is the scattering rate of reservoir excitons into the condensate, $P(\mathbf{r})$ is the non-resonant pump(s), $\gamma$ is the rate of losses of condensed polaritons through the cavity mirrors, $\Gamma$ is the rate of redistribution of reservoir excitons between the different energy levels, and $\Lambda$ is a phenomenological energy relaxation parameter[67] (simulation parameters can be found in SI, Section I).

We start by investigating stationary states characterised by in-phase and anti-phase bonding between condensates in the absence of frustration. Figure 1d–f show experimental results for real space, Fourier space ($k_x$–$k_y$ momentum plane), and phase of the polariton photoluminescence (PL) in a hexagon of radius $R = 16.9\,\mu m$. The coupling is clearly distinguishable as $J > 0$ from the odd number of bright fringes between the vertices, evidencing in-phase locking. Figure 1j–l show same measurements for a hexagon of radius $R = 14.7\,\mu m$, where we now observe an even number of fringes between the vertices ($J < 0$), evidencing anti-phase locking. Figure 1g–i and m–o show corresponding simulations of the steady state, in-phase, and anti-phase locked, polariton condensate wavefunction using the driven-dissipative Gross–Pitaevskii equation in agreement with experimental observations. Other polygon geometries for both in-phase and anti-phase configurations of the condensates are given in the SI, Section II.

We investigate the system where we time-integrate over multiple instances of the condensate. Here, we are unable to directly observe the frustrated current in one direction, or the other, due to the stochastic choice of chirality during condensation and so relative condensate phase readout through interferometric techniques averages out. In-phase and anti-phase states with $\theta_{i,i+1} = 0$ or $\pi$ are the notable exception, which are observable in averaged measurements because they are non-degenerate. Nevertheless, as we describe in the following, two stochastically appearing counter-circulating currents ($\pm m$) give rise to a spatially oscillating coherence pattern that allows us to uniquely determine the vortex state $|m|$ in averaged measurements. The first-order mutual coherence function,

$$g_{ij}^{(1)} = \frac{\langle \Psi_i^* \Psi_j \rangle}{\sqrt{\langle |\Psi_i|^2 \rangle \langle |\Psi_j|^2 \rangle}} \qquad i, j \in \{1, \dots, N\}, \quad (3)$$

describes the coherence properties between two condensates $i$ and $j$ with wavefunctions $\Psi_{i,j}$ in a polygon of $N$ coupled condensates. In the following, we assume that nodes are labelled sequentially $1, 2, \dots, N$ along the polygon. For a circulating current of charge $m$ along the polygon satisfying $\langle |\Psi_i|^2 \rangle = \langle |\Psi_j|^2 \rangle$, the complex coherence function is given by $g^{(1)}(d) = \exp(i2\pi md/N)$, where $d = |i - j|$. In the case of equal probabilities of the two stochastically appearing degenerate counter-propagating currents with charge $\pm m$ the mutual coherence function becomes a standing wave,

$$g^{(1)}(d) = \cos(2\pi md/N). \quad (4)$$

By extracting the mutual coherence function $g^{(1)}(d)$ along the polygon nodes, we are able to determine the charge $|m|$ (or alternatively Bloch mode) of the stochastically appearing degenerate currents. In Fig. 2c, we show the measured (red crosses) values of the mutual coherence function for the systems shown in Fig. 2a, b. The first column displays the in-phase hexagon with all relative phases at zero and a high coherence. The second column shows an anti-phase hexagon again with a high coherence but now interchanging between $+1$ and $-1$, reflecting the phase jumps along the polygon in the anti-phase configuration. The black dots in the coherence graphs indicate the value calculated by Eq. (4) and the green dashed line represents the $g^{(1)}$ function replacing $d$ for a continuous variable.

The most interesting phase configuration is predicted for an odd number of condensates in the frustrated regime ($J < 0$), shown for a heptagon in the final three columns with the fourth column depicting a phase difference of $\theta_{i,i+1} = \pm 2\pi/7$ indicating a vortex of $|m| = 1$, the fifth column has $\theta_{i,i+1} = \pm 4\pi/7$ for a vortex of $|m| = 2$, and the final column has $\theta_{i,i+1} = \pm 6\pi/7$ for a vortex of $|m| = 3$. The real values of the $g^{(1)}$ are extracted from interferograms between the selected condensates with $d = 0$ representing the interference of a condensate with itself. We note that the in-phase heptagon ($m = 0$) is displayed in the third column and has a high coherence between all condensates. The coupling $J$ in the polygon is controlled by altering the radius of the polygon, which we detail in the SI, Section III. The extracted data (red crosses) is in good agreement with the predicted values (black dots). It is instructive to compare the extracted coherence properties of the anti-phase state of the hexagon with the frustrated states of the heptagon. In the former, clear switching of $g^{(1)}$ between $-1$ and $1$ can be seen, which gives a total of three full periods around the hexagon, while for the circulating states with $\theta_{i,i+1} = \pm 2\pi/7$, $\pm 4\pi/7$, $\pm 6\pi/7$ we observe one, two, and three total periods around the heptagon, respectively. We also note that for the circulating states in the heptagon the discrete values do not occur at the peak or trough of the wave, except for the case of $d = 0$ but at intermediary values due to the averaged phase of the circulating state. These results evidence nontrivial phase configuration appearing in the condensate polygon, which can be attributed to frustration and spontaneous onset of a circulating polariton current. We point out that our results are in agreement with the ensemble averaged measurements performed on lattices of coupled laser systems[68].

We provide further evidence on the formation of a circulating polariton current by investigating condensation in the single shot regime where a single instance of the condensate polygon is excited and measured. Figure 3a shows the experimental real-space PL of a frustrated pentagon ($N = 5$), wherein we observe an even number of interference fringes between neighbouring condensates. Figure 3b shows the Fourier-space PL where we observe an absence of radial nodal lines in contrast to Fig. 1k. We point out that the presence of radial nodal lines in the Fourier space PL would evidence anti-phase locked condensates, corresponding to

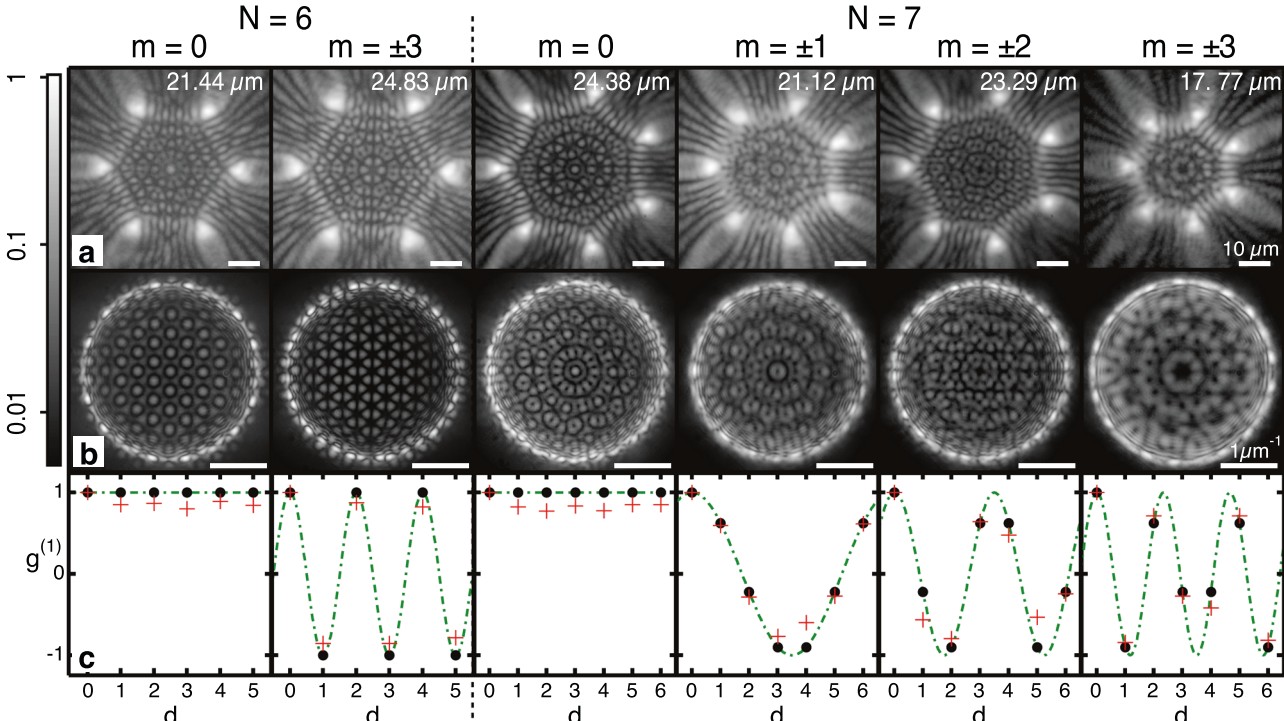

**Fig. 2 Extraction of the $g^{(1)}$ from stationary and circulating states in both a hexagon and heptagon. a, b** Real- and Fourier-space PL imaging of a hexagon and heptagon in different coupling configurations achieved by tuning the polygon radius. Two leftmost columns show condensation of a hexagon into in-phase ($m = 0$) and anti-phase ($m = 3$) configurations, whereas four rightmost columns show in-phase ($m = 0$) and vortex formation ($m = \pm 1$, ±2, ±3) of a heptagon. **c** Extracted (red crosses) and calculated (black dots) mutual first-order coherence $g^{(1)}$ versus parameter $d$ extracted from the interferogram between condensates for different polygon configurations. The values of $g^{(1)}$ lie upon the continuous function (green dot-dashed line) given by $g^{(1)}(x) = \cos(2\pi mx/N)$, where $x$ has replaced the discrete parameter $d$. The point at $d = 0$ corresponds to the autocorrelation of the condensate wavefunction at zero time delay and is set as $g^{(1)}(0) = 1$. The radii of the polygons are written in the top left hand corner of the real-space images in **a**. All real- and Fourier-space images are plotted on the same scale defined on the scale bar at the end of each row.

standing wave formation along the polygon and a zero-net cur-rent. Here, however, the lack of nodal lines suggests the presence of a net particle current along the polygons edge. For comparison, we show numerical simulations in Fig. 3e, f of the real- and Fourier-space density of the condensate wavefunction resulting in a stable discrete vortex solution with a winding $m = -1$. We note that the winding number sign in simulation is stochastically determined from random initial conditions.

Although the frustrated vortex regime can be classified from the nontrivial interference patterns appearing in the real- and Fourier-space PL, it does not concretely verify the presence of a vortex state. We therefore resolve the spatial phase distribution of the vortex using off-axis digital holography[43] (see "Experimental techniques" in "Methods"). During condensation, the vortex formation stochastically results in either clock- or anticlock-wise winding of the phase along the polygon edge. Averaging over several condensate instances would therefore skew direct mea-surement of the phase. Instead, in order to view the vortex state, we utilise a single condensate realisation detection scheme, whereby each image of the polygon is the result of a single instance of the condensate formation. Figure 3c shows the experimental phase of the condensates with a $m = -1$ vortex, extracted from its interferogram, where it can be seen to spiral about the polygon's centre. Figure 3d shows a zoomed-in region corresponding to the black dashed square in Fig. 3c. Figure 3g, h show a corresponding real-space phase map from the simulation in Fig. 3e, f. To further verify that the entire condensate possesses a circulating current, we directly measure the phase coming from the bright emission spots at the polygon vertices where most of

the condensate resides. The experimentally measured relative phase between neighbouring condensates is demonstrated with black dots in Fig. 3i, which are extracted from the regions denoted by black circles in Fig. 3c. The relative phases are close to the expected vortex value of $\theta_{i,i+1} = 2\pi/5$ (red line) with a maximal deviation of 0.6 radians (blue and green dash-dotted lines). Line profiles taken along the black circles in Fig. 3d, h are shown in Fig. 3j and unveil a full $2\pi$ phase winding about the vortex core, where the experimental data are represented by the coloured disks and the numerical simulation by the black-dotted curve.

In Fig. 4, we show an experimental realisation of a frustrated pentagon condensate containing a vortex of charge $m = 2$. Figure 4a shows the experimental real-space PL wherein we again observe an even number of interference fringes between neighbouring condensates in real space. Figure 4b, c show the phase of the polygon condensate, which can be seen to wind about its centre characterised by two phase singularities at the core. Figure 4d–g show a corresponding numerical simulation of a steady-state $m = 2$ vortex. In Fig. 4h, we show the experimentally measured relative phase between neighbouring condensates (black dots), extracted from the regions denoted by black circles in Fig. 4b, revealing the expected double vortex phase difference of $\theta_{i,i+1} = 4\pi/5$ (red line) with a maximal deviation of 0.45 radians (blue and green dash-dotted lines). Line profiles taken along the black circles in Fig. 4c, f now unveil a full $4\pi$ phase winding about the vortex core (see Fig. 4i). We observe that the line profile of the simulation (black-dotted curve, Fig. 4i) is not straight but has small oscillations along the azimuthal angle. We attribute this to the

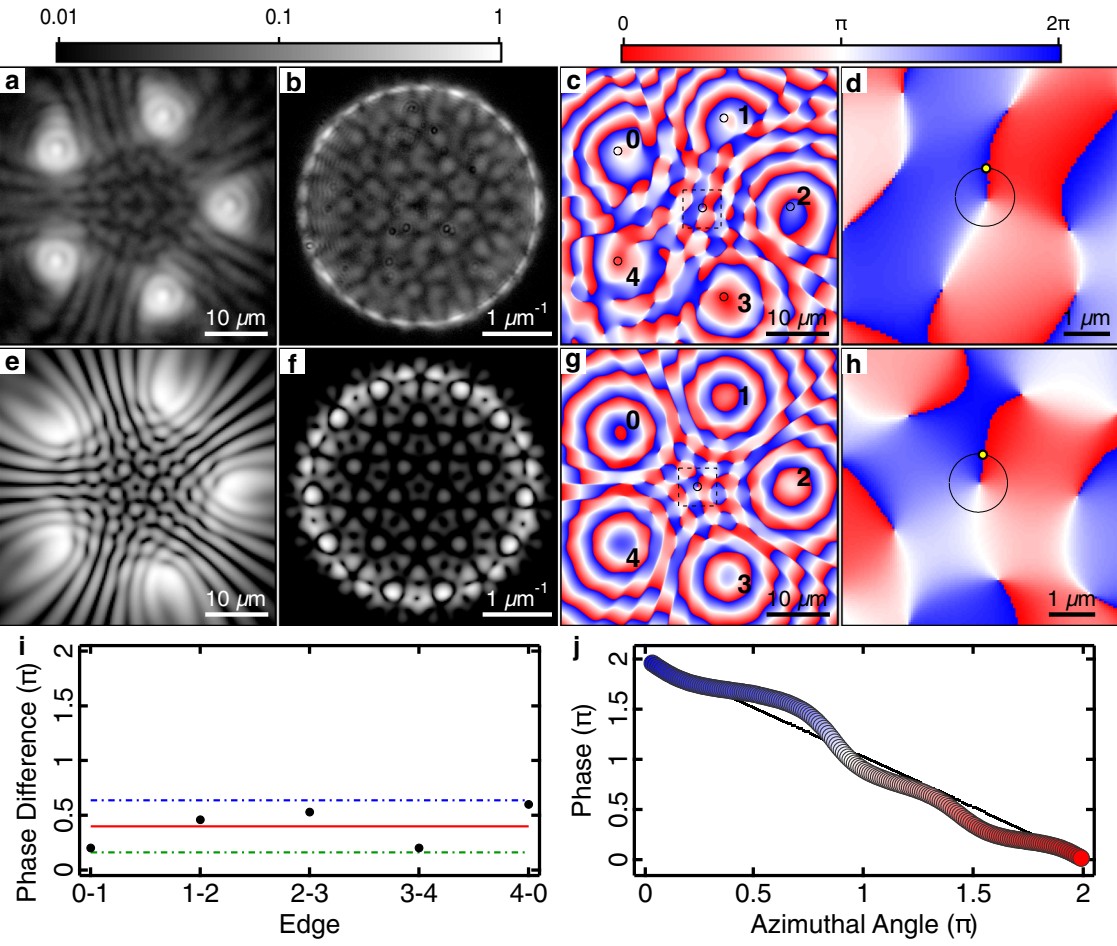

**Fig. 3 Single shot of a pentagon of polariton condensates with a singly charged vortex at the centre.** Pentagon condensate with a charge $m = -1$ vortex. **a** Experimental real-space PL, **b** Fourier-space PL and **c**, **d** phase extracted from interferometry from a single instance of the condensate. Simulated steady state $m = -1$ vortex wavefunction **e** real-space density, **f** Fourier-space density, and **g**, **h** phase. **d**, **h** Zoomed-in regions of the dashed squares in **c**, **g**, respectively. **i** Phase difference between adjacent condensates (black dots) around the polygon extracted from the black circles in **c**. The red line is the expected phase difference of $\theta_{i,i+1} = 2\pi/5$. Green and blue dash-dotted lines mark the maximum deviation. **j** Line profile extracted along the black circle in **d**, **h** demonstrating a full $2\pi$ phase winding. Yellow dots correspond to azimuthal angle equal to zero. Experimental data are represented by the coloured disks and numerical simulation by the black-dotted curve.

system not being cylindrically symmetric, and so angular momentum is no longer a good quantum number and the 2D angular harmonics $e^{iw\varphi}$ should be replaced by angular Bloch solutions $u_{q,n}(r, \varphi)e^{iq\varphi}$, which in general possess more complex phase structure leading to the extra modulation of the phase seen in simulations, similar to what has been seen for vortices in optical lattices like in ref. [26]. We note that we do not externally imprint the vortex phase[69,70], but we rather control the dynamics of the coupling between condensates, which leads to the spontaneous formation of these vortices. In Section IV in the SI, we estimate the probability of vortex formation through simulations of varying polygon radii, and irregularities.

We observe that in some instances for frustrated odd-numbered polygons, the condensates do not successfully "agree" with each other on which direction to form a current. This leads to a state that does not have a single central vortex but rather forms a vortex–antivortex pair as shown in Fig. 5. Figure 5a, b shows the experimental real-space magnitude of the PL and phase, respectively. The diagonal dashed lines are guide to the eye marking a split in density across the condensate polygon. Figure 5c shows a zoomed-in region corresponding to the black dashed square. A vortex–antivortex pair can be observed with

cores separated by ~1 μm. Figure 5d–g shows a corresponding numerical simulation of a condensate steady state with a vortex–antivortex pair, obtained by displacing pump spot number 4 in the pentagon by 2.8 μm radially outwards. Figure 5h shows that the measured phase difference between the vertices 0–1, 1–2, 2–3, and 0–3 is equal to $\pi$, whereas the phase of condensate 4 is not well defined due to destructive interference, fracturing it into two parts. Line profiles taken along the black circles in Fig. 5c, f unveil an opposite $2\pi$ phase winding about each of the vortex cores (see Fig. 5i).

## Discussion

We lastly discuss some sources of discrepancies appearing between theory and observations. We first underline that the mean-field model [Eqs. (1) and (2)] assumes ideal and symmetric conditions, whereas experiment is always subject to some error in the position and intensity of the excitation spots, and additionally sample disorder that cannot be avoided. Second, we have only included a phenomonological energy relaxation mechanism $\Lambda$ for simplicity, whereas better agreement between observations and theory can possibly be achieved by including directly the

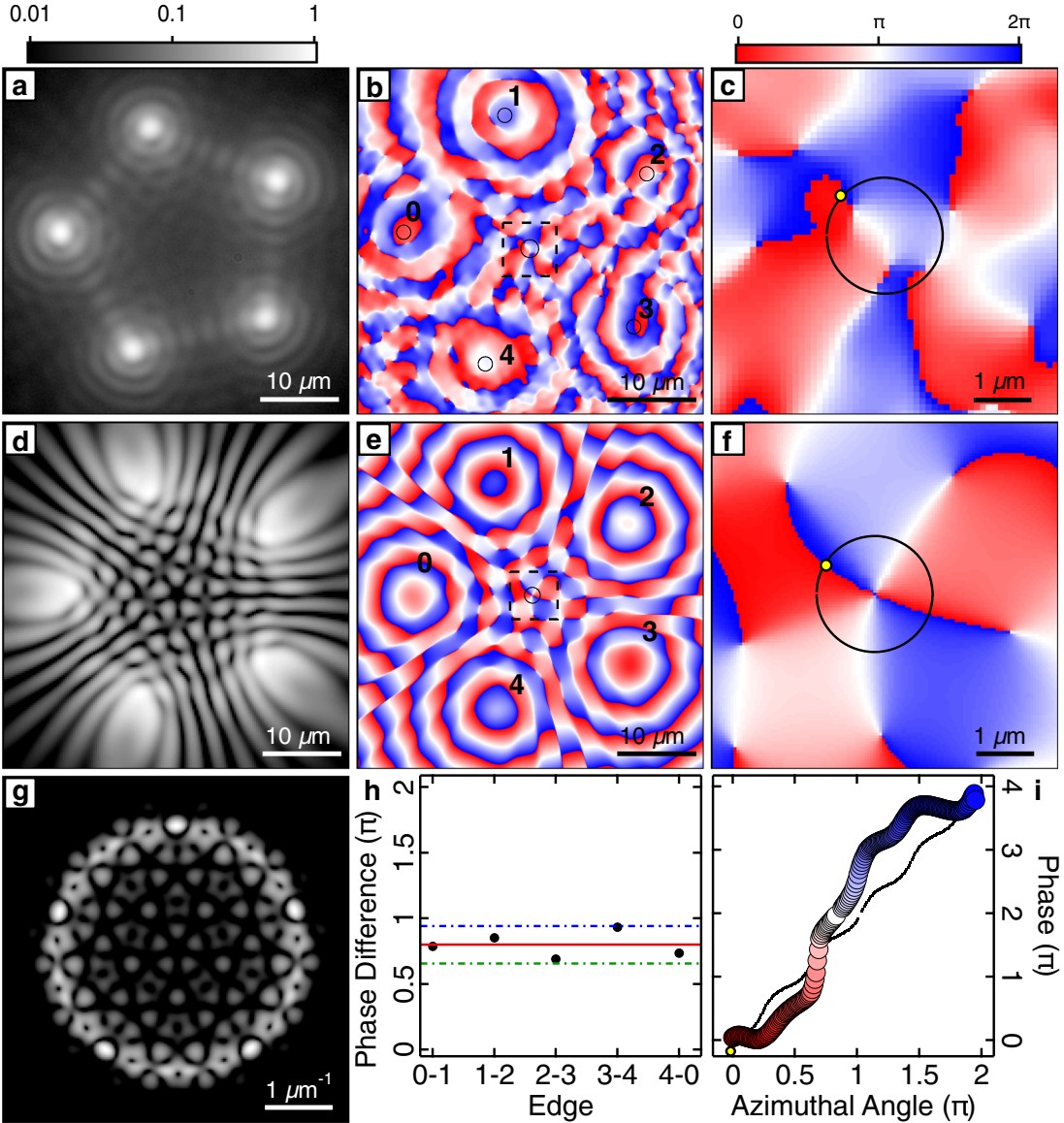

**Fig. 4 Single shot of a pentagon of polariton condensates with a doubly charged vortex at the centre.** Pentagon condensate with a charge $m = 2$ vortex. **a** Experimental real-space PL and **b**, **c** phase extracted from interferometry from a single instance of the condensate. Simulated steady state $m = 2$ vortex wavefunction **d** real-space density, **g** Fourier-space density, and **e**, **f** real-space phase. **c**, **f** Zoomed-in regions of the dashed squares in **b**, **e**, respectively. **h** Phase difference between adjacent condensates (black dots) around the polygon extracted from the black circles in **b**. The red line is the expected phase difference of $\theta_{i,i+1} = 4\pi/5$. Green and blue dash-dotted lines mark the maximum deviation. **i** Line profile extracted along the black circle in **c**, **f** demonstrating a full $4\pi$ phase winding. Yellow dots correspond to azimuthal angle equal to zero. Experimental data are represented by the coloured disks and numerical simulation by the black-dotted curve.

dynamics of the exciton reservoir into the energy relaxation of the polaritons[71].

Our experimental and numerical observations provide strong evidence of the presence of stable polariton currents resulting in discrete vortices of winding numbers $|m| \geq 1$ appearing spontaneously in odd numbered polygon structures of exciton–polariton condensates. Controlling the size of the polygon allows one to tune the coupling between condensates from being in-phase locked to anti-phase locked. In the latter, for an odd numbered polygon, the combined effects of antibonding frustration and parity symmetry breaking leads to the formation of circulating polariton currents along the polygon's edge, causing the presence of vortices at the centre with winding number $|m| \leq (N-1)/2$ where $N$ is the number of condensates (vertices) in the polygon. The future outlook for these circulating currents can involve tuning the geometry and the profile of the excitation beams so as to

effectively control the amount of the circulating polariton fluid, its tangential and inward radial velocities fields, and even the density of the condensate residing in the central region of the polygon around the core of the vortex, thus providing an flexible platform to study light–matter vorticity.

## Methods

**Sample.** The semiconductor microcavity structure studied here is a planar, strain-compensated $2\lambda$ GaAs microcavity with embedded InGaAs QWs. Strain compensation was achieved by $AlAs_{0.98}P_{0.02}$/GaAs distributed Bragg reflector layers instead of the thin AlP inserts in the AlAs layers used in ref. [72] as their effective composition could be better controlled. The bottom distributed Bragg reflector consists of 26 pairs of GaAs and $AlAs_{0.98}P_{0.02}$ while the top has 23 of these pairs, resulting in very high reflectance (>99.9%) in the stop-band region of the spectrum. The average density of hatches along the [110] direction was estimated from transmission imaging to be about 6/mm, while no hatches along the [1$\bar{1}$0] direction were observed. Three pairs of 6 nm $In_{0.08}Ga_{0.92}As$ QWs are embedded in the GaAs cavity at the anti-nodes of the field as well as two additional QWs at the first and last node

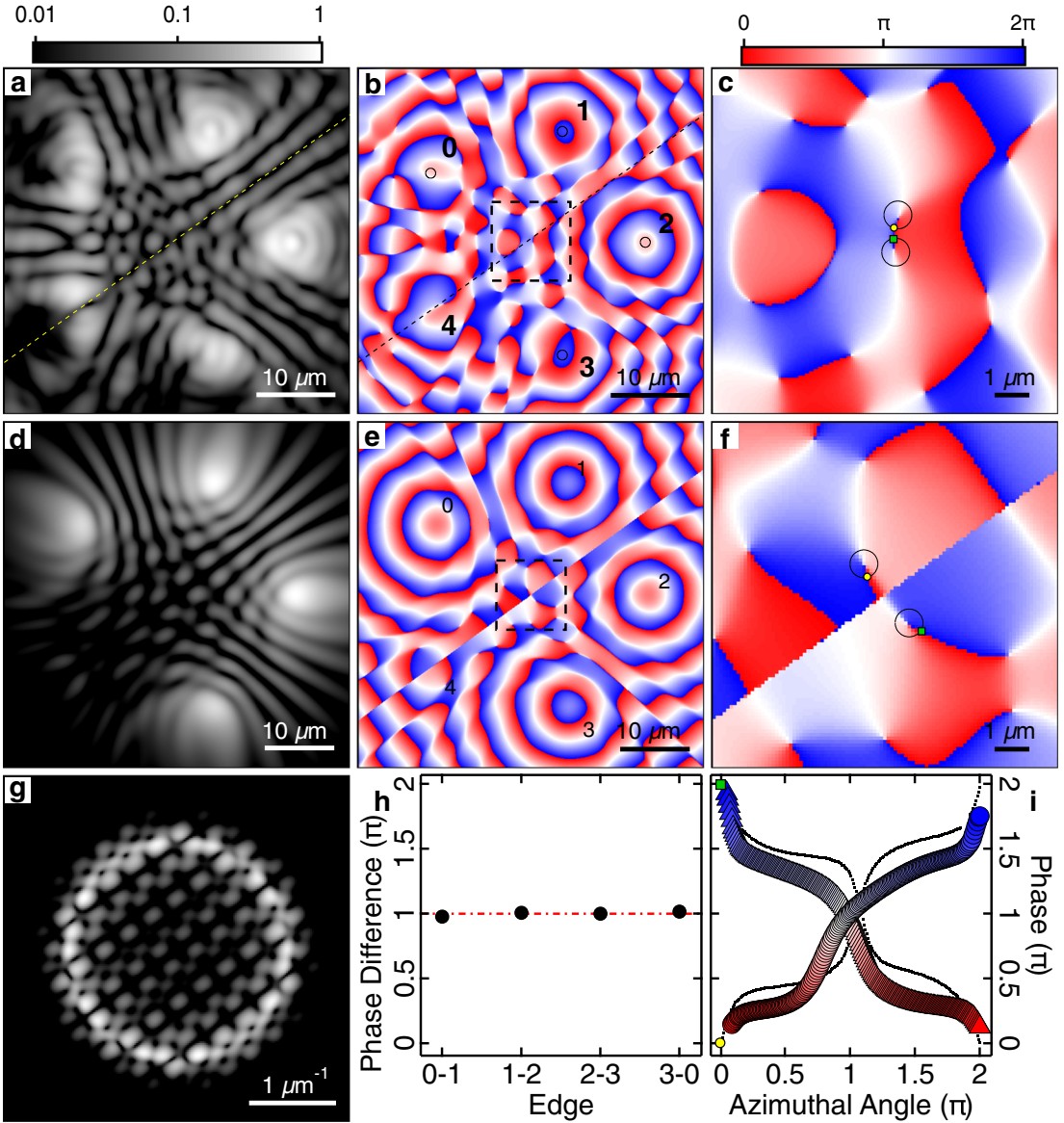

**Fig. 5 Single shot of a pentagon of polariton condensates with one fractured condensate.** Pentagon condensate with a $|m| = 1$ vortex–antivortex pair. **a** Experimental real-space PL and **b**, **c** phase extracted from interferometry from a single instance of the condensate. Simulated steady-state vortex–antivortex wavefunction **d** real-space density, **e**, **f** phase and **g** Fourier-space density. **c**, **f** Zoomed-in regions of the dashed squares in **b**, **e**, respectively. **h** Phase difference between different condensates (black dots) in the polygon extracted from the black circles in **b**. **i** Line profiles extracted along the two dashed circles in **c**, **f**. Yellow and green markers correspond to azimuthal angle equal to zero. Experimental data are represented by the coloured disks and numerical simulation by the black-dotted curves. Image **a** is saturated at a lower limit of 0.04 to allow for better visibility of the fringes.

to serve as charge carrier collection wells. The large number of QWs was chosen to increase the Rabi splitting and keep the exciton density per QW below the Mott density[73] and also for obtaining sufficiently high polariton densities to achieve polariton condensation under non-resonant excitation. The strong coupling between the exciton resonance and the cavity mode is observed with a vacuum Rabi splitting of $2\hbar\Omega \sim 8$ meV. A wedge in the cavity thickness allows access to a wide range of exciton-cavity detuning. All measurements reported here are taken at detuning $\Delta \approx -5.5$ meV. The measured $Q$-factor is ~12,000, while the calculated bare cavity $Q$-factor, neglecting in-plane disorder and residual absorption, is ~25,000. As the emission energy of the InGaAs QWs is lower than the absorption of the GaAs substrate, we can study the PL of the sample both in reflection and transmission geometry. The transmission geometry, which is not available for GaAs QWs, allows us to filter the surface reflection of the excitation and has been widely utilised to probe the features of polariton fluids[74,75] under resonant excitation of polaritons. Using real- and Fourier-space imaging under non-resonant optical excitation, polariton condensation, and a second threshold marking the onset of photon lasing, the transition from the strong- to the weak-coupling regime was studied in this microcavity[64].

**Experimental techniques**. In the experiments described here, the sample was held in a cold finger cryostat at a temperature of $T \approx 4$ K. CW excitation is provided by a single-mode Ti:Sapphire laser modulated by an acousto-optic modulator (AOM) to form a quasi-CW beam. We use non-resonant excitation from the epi side and utilise two different detection schemes: detect the emission from the substrate side so that the excitation is filtered by the absorption of the GaAs substrate (transmission geometry) or detect the emission from the epi side to prevent disruption of the Fourier-space imaging (reflection geometry). The optical excitation, for all the measurements reported in this work, is at first reflectivity minimum above the cavity stop band. The spatial profile of the excitation beam is modulated to a regular polygon with Gaussian profiles at each vertex of approximately equal in diameter spots using a reflective SLM. We use high numerical aperture microscope objectives ($0.4 \le$ NA $\le 0.5$) to focus the spatially modulated beam to ~1–2 μm in diameter, at full width at half maximum, excitation spots and to collect the PL in reflection geometry. The PL from the sample can also be collected in transmission geometry with ±25° collection angle, by a 0.42 NA microscope objective. The resonant seed laser (Toptica) mode is first cleaned with pinhole to ensure a clean phase front. Then the seed laser is split into two paths at PBS1 with a small amount

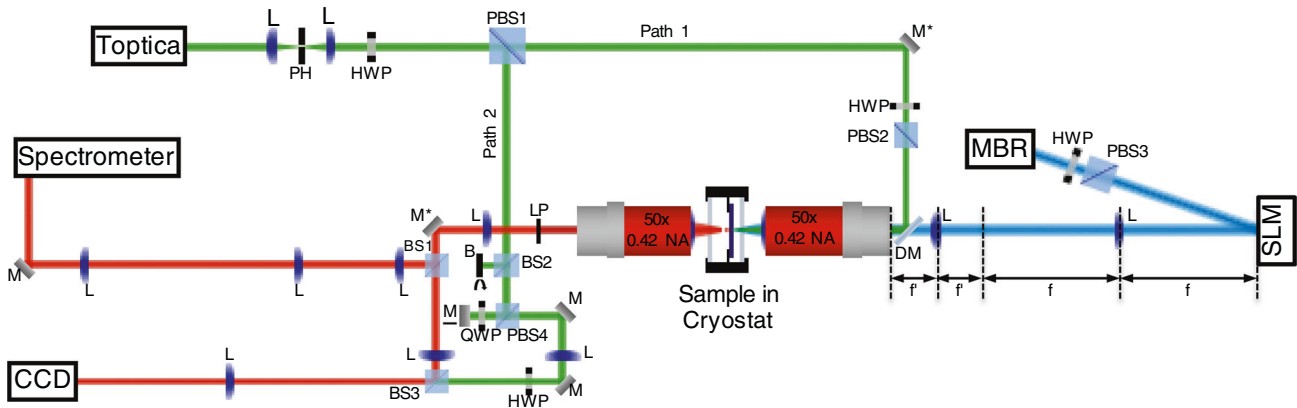

**Fig. 6 Experimental set-up.** Equipment: mirrors (M, M* (periscope), M (mounted on translation stage), DM (dichroic mirror)); lenses (L); polarising beam splitter (PBS); non-polarising beam splitter (NPBS); half-wave plate (HWP); quarter-wave plate (QWP); long-pass filter (LP); and beam block (B). Simultaneously, real-space PL is detected onto the uncoupled CCD and 2D Fourier space is projected onto the spectrometer allowing us to view either the 2D Fourier space or the energy-resolved dispersion by using the grating. The external seed laser is brought to be resonant with the PL by going on the grating and detecting the PL from BS1 and the laser by removing block (B) in front of BS2 and tuning the external laser. For the homodyne interferometry, the external seed laser is split before the sample, along path 1, a small amount seeds one of the condensates, while along path 2, the remainder is interfered with the PL on BS3. The path of the external seed laser is shown in green, the PL is shown in red and the excitation laser is shown in blue.

down path 1, directed onto the sample using a dichroic mirror, which passes the excitation laser but reflects the PL wavelength; and the rest down path 2 where the laser is interfered with the PL on BS3 for homodyne interferometry. The remainder of the excitation laser is filtered with a long pass filter and the paths are brought to zero time delay by controlling mirror M, which is on a linear translational stage. 2D Fourier- and real-space PL imaging are obtained by projecting the corresponding space onto a cooled charge-coupled device (CCD) camera.

Initially, we viewed the system by time averaging over multiple instances of the condensate to obtain real- and Fourier-space images. The PL is then directed at a second SLM (not shown) where the hologram design allowed us to investigate the first-order mutual coherence function between any two condensates from the polygon[76]. The selected PL was projected onto a CCD in Fourier space so that the mutual coherence between the two condensates under investigation could be seen in the resultant interferogram. The magnitude of the $|g^{(1)}|$ is extracted from the interferograms by fitting with

$$I_{\text{int}} = I_1 + I_2 + 2\sqrt{I_1 I_2}|g^{(1)}|\cos(kx + \theta),\qquad(5)$$

where $I_{\text{int}}$, $I_1$, and $I_2$ are read from line profiles along a direction $x$ taken across the images of the interference, reference arm 1, and reference arm 2, respectively[77]. The fitting parameters are the phase ($\theta$), wavenumber ($k$), and coherence magnitude ($|g^{(1)}|$), allowing us to directly extract the phase and coherence for each data set.

Since the phases across the polygon cannot be measured directly, we utilise homodyne interferometry[78] (see Fig. 6). We seed one of the condensates with part of an external laser (Toptica), which is tuned to be resonant with the condensate. The external seed perturbs the condensate to the frequency of the seed laser, by utilising the U(1) symmetry breaking at the point of condensation, without interfering with the dissipative dynamics of the condensate. The PL is then interfered with the remainder of the external seed laser on the third non-polarising beam splitter (BS3). We then use off-axis digital holography[43] to extract both magnitude and phase of the condensate order parameter $\Psi$ at the resonance energy $\hbar\omega$ of the seed laser. Furthermore, to view a single instance of the condensate, we trigger the AOM once, thereby creating a single instance of the condensate, which is 20–60 µs in length. We synchronise the CCDs detecting real space, Fourier space, and energy-resolved dispersion to the AOM thereby ensuring that the CCDs record the entirety of each instance of the condensate.

## Data availability
All data supporting this study are openly available from the University of Southampton repository at https://doi.org/10.5258/SOTON/D1738.

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

## Acknowledgements

T.C., H.S., J.D.T., M.S. and P.G.L. acknowledge the support of the UK's Engineering and Physical Sciences Research Council (grant EP/M025330/1 on Hybrid Polaritonics). H.S. and P.G.L. acknowledge support by the European Union's Horizon 2020 program, through a FET Open research and innovation action under the grant agreement No. 899141 (PoLLoC). S.A. and P.G.L. acknowledge the support by RFBR according to the research project No. 20-52-12026 (jointly with DFG). N.G.B. acknowledges the support from Huawei. K.K. acknowledges the support from EPSRC and Cambridge Trust.

## Author contributions

N.G.B. and P.G.L. designed the research. T.C., J.D.T., S.A., M.S. and P.G.L. performed the experiments and analysed the experimental data. W.L. and P.G.L. contributed to the design and fabrication of the microcavity. H.S. contributed on the numeric simulations and Bloch analysis. K.K. and N.G.B. contributed on the theoretical calculations. The manuscript was written through contributions from all authors. All authors have given approval to the final version of the manuscript.

## Competing interests

The authors declare no competing interests.
