## [Peer Review File · Nature Communications]

Reviewers' Comments:

Reviewer #1:

Remarks to the Author:

The authors study in great detail and present very nicely the creation of vortices of different topological charge in Bose-Einstein-like condensates realised with polaritons in semiconductor microcavities. This is a difficult task that has been achieved only in a few cases by imprinting the vortices and leading to unstable vortices. In this case, stable discrete vortex states with large topological charge are attained resulting from antibonding frustration between nearest neighbouring condensates.

These are novel and very interesting results for the Physics community and in particular for the Condensed Matter and Atomic Physics community which deserve publication. The work is of great significance and very original and provides new ways to study light-matter vorticity. The manuscript is very clearly written, they provide a nice review of the state of the art, furthermore, the experimental data are of utmost quality supporting undoubtedly the conclusions and claims.

Therefore, I strongly support publication in Nature Communications.

There are only some minor points that could be considered by the authors in order to improve the quality of the presentation, namely, and listed by order of appearance in the manuscript:

- a)Page 3: References 33 & 34 seem to be swapped.
- b)Page 3: the adverb "currently"; is used, and the references seem to be a bit old for this (years 2008 - 2014). Also "recently", in page 6 (2016).
- c)Page 4-5: It should be mentioned somewhere here in the text that the scales, spatial and false colour are presented in VI. IMAGE PARAMETERS of the Supplementary Information, and not wait that the reader gets to this section to learn about which scales are used in the figures.
- d)Page 10, Fig. 3(j): it is not evident that the marked fork dislocation highlighted by the yellow line is the only one present in this inset. Same comment for Fig. 4(b). Could the authors comment on that?
- e)Page 11, Fig 4(j): while in Fig. 3(j) one can follow the experimental data going around the black circle in 3d, it is not very clear how the experiments are plotted in Fig 4(j) circling with the profile shown in 4d.
- f)In the same figure as in e), could the authors comment on the oscillations seen in the phase of the simulations?
- g)Page 13, Fig. 5(b): Difficult to see the fork dislocations in the figure, maybe the yellow lines difficult to see them more than helping to do it.
- h)Supplementary information, page 2 and Fig. S1: acronyms are not fully defined.
- i)Supplementary information, page 2 and Fig. S1: the numerical aperture of the focusing objective is specified as 0.42 in the text while in the figure appears as 0.5.
- j)Supplementary information, page 3, it is written: "...where the hologram design allowed us to investigate the relationship between any two condensates...";. To which relationship are referring the authors to? To the phase relationship?
- k)Supplementary information, page 3, it is written: "...and the extracted real-space magnitude of the polygon using..." What is meant by magnitude of the polygon?
- l)There is a work also on polariton condensates about the influence of particle flow on the appearance of topological defects which maybe is relevant for the subject presented here and that is not quoted, namely the work in Optics Express **20**, 16366 (2012).

Reviewer #2:

Remarks to the Author:

The paper by Cookson et al examines the vortices which can be realised in lattices of non-resonantly pumped polariton condensates. They find that, in cases where geometric frustration is present, complex vortices with non-trivial winding numbers can be observed and thoroughly characterised.

They show how the average behaviour is well described by a simple GPE model and also that the representative physics of single shot measurement can be obtained from the same calculations. The experiments undertaken here are really beautiful and the level of control shown is impressive. The results represent a significant improvement over previous similar experiments showing much finer control over a larger lattice. This is a valuable step towards utilising these types of device for analogue computing applications. With these points in mind I recommend publication in Nature Communications.

There are a few minor points which the authors may want to consider before publication:

The discuss of the model they use is entirely in the supplementary information. It would be useful to at least have the GPE in the main text so that readers can easily see what effects have been included

It would be useful to have slightly more discussion about the discrepancies between the experiment and the theory results. Some of this is really just to do with the stochastic nature of sthe observations, but some is more systematic, for example the asymmetry and slight disagreement in the experimental results of Figs 3,4,5 (j).

Response to reviewer 1:

Report on NCOMMS-20-18055A-Z: Geometric frustration in polygons of polariton condensates creating vortices of varying topological charge, by Tamsin Cookson et al.

The authors study in great detail and present very nicely the creation of vortices of different topological charge in Bose-Einstein-like condensates realised with polaritons in semiconductor microcavities. This is a difficult task that has been achieved only in a few cases by imprinting the vortices and leading to unstable vortices. In this case, stable discrete vortex states with large topological charge are attained resulting from antibonding frustration between nearest neighbouring condensates.

These are novel and very interesting results for the Physics community and in particular for the Condensed Matter and Atomic Physics community which deserve publication. The work is of great significance and very original and provides new ways to study light-matter vorticity. The manuscript is very clearly written, they provide a nice review of the state of the art, furthermore, the experimental data are of utmost quality supporting undoubtedly the conclusions and claims. Therefore, I strongly support publication in Nature Communications.

There are only some minor points that could be considered by the authors in order to improve the quality of the presentation, namely, and listed by order of appearance in the manuscript:

a) Page 3: References 33 & 34 seem to be swapped.

Thank you, these have now been swapped round. They are highlighted in the references to make it clear they have been switched and are now references [35] & [34] respectively.

b) Page 3: the adverb "currently"; is used, and the references seem to be a bit old for this (years 2008 - 2014). Also "recently", in page 6 (2016).

Page 3: has been altered to "have been the subject of intense investigation for some time" since the references in this section and throughout the paper agree with this.

Page 6: changed "recently" to "in 2016"

Page 3: In addition, to this end, an instance of "recently" was also removed here as the reference was for 2016 (Baboux, et al. PRL, 2016).

c) Page 4-5: It should be mentioned somewhere here in the text that the scales, spatial and false colour are presented in VI. IMAGE PARAMETERS of the

Supplementary Information, and not wait that the reader gets to this section to learn about which scales are used in the figures.

This is currently mentioned in the figure 1 caption, but an additional sentence has been added to the text to make it clear this is where all the information is: “We note that *all image parameters are given in SI, Section VI*” on page 8 at the end of the paragraph where Fig. 1 is discussed.

d) Page 10, Fig. 3(j): it is not evident that the marked fork dislocation highlighted by the yellow line is the only one present in this inset. Same comment for Fig. 4(b). Could the authors comment on that?

We thank the referee for this valuable comment! Indeed, the central region of the polygon possesses very little light which makes it challenging to resolve the position of fork-like dislocations in the interferogram. This is precisely the reason that we measure the perimeter of the polygon where the condensate is the strongest revealing the rotation of the phase, which is the main point and novelty of the paper. Scrutinising the centre of the polygon is not necessary to verify the system vorticity. None-the-less, we have chosen to present the zoomed in phase maps in Figs.3(d), 4(c), and 5(c) to illustrate what our measurements reveal in the polygon centre despite the lack of light, which agrees surprisingly well with our model. We have now removed the zoomed interferogram plots from Figs.3-5 and focus instead on the phase maps for clarity.

e) Page 11, Fig 4(j): while in Fig. 3(j) one can follow the experimental data going around the black circle in 3d, it is not very clear how the experiments are plotted in Fig 4(j) circling with the profile shown in 4d.

We have now added coloured markers on the black circles in Figs. 3(d,h), 4(c,f) and 5(d,h) that indicate the position of the zero azimuthal angle in the horizontal axis of the corresponding line profile plots.

f) In the same figure as in e), could the authors comment on the oscillations seen in the phase of the simulations?

Since the system doesn't have cylindrical symmetry on has that angular momentum is no longer a good quantum number and the 2D angular harmonics $e^{i\ell\varphi}$ should be replaced by angular Bloch solutions $u_{q,n}(r, \varphi)e^{iq\varphi}$ which in general possess more complex phase structure leading to the extra modulation of the phase seen in simulations, similar to what has been seen for vortices in optical lattices like in Ref.[26].

g) Page 13, Fig. 5(b): Difficult to see the fork dislocations in the figure, maybe the yellow lines difficult to see them more than helping to do it.

Please see our answer to comment (d).

h) Supplementary information, page 2 and Fig. S1: acronyms are not fully defined. The acronyms have now been fully defined in the figure caption.

i) Supplementary information, page 2 and Fig. S1: the numerical aperture of the focusing objective is specified as 0.42 in the text while in the figure appears as 0.5. There are several different versions of the set up and excitation scheme described here used throughout this work: only Figure 4 used a 0.5NA 100x objective for excitation, Figures 1,3,5,S2,S3 used a 0.42NA 50x objective for excitation and Figure 2 used a 0.4NA 20x objective for excitation. We now give a range of excitation numerical apertures and in the figure the objective is labelled with the one used in the highest number of figures. The alteration in the text is amended to:

“We use high numerical aperture objectives ($0.4 \leq NA \leq 0.5$) to focus...”

During this review we also felt it necessary to make a clear distinction between the excitation laser, seed laser and PL. Therefore, the colour of the seed laser has been altered to make the figure clearer between excitation laser and the figure S1 caption amended to include the following sentence:

“The path of the external seed laser is shown in green, and the PL is shown in red, and the excitation laser is shown in blue.”

And we also added in more detail explaining the set up to the description on page 3 of the SI:

“...The resonant seed laser (Toptica) mode is first cleaned with pinhole (PH) to ensure a clean phase front. Then the seed laser is split into two paths at PBS1 with a small amount down path 1, directed onto the sample using a dichroic mirror (DM) which passes the excitation laser but reflects the PL wavelength; and the rest down path 2 where the laser is interfered with the PL on BS3 for homodyne interferometry. The remainder of the excitation laser is filtered with a long pass filter (LP) and the paths are brought to zero time delay by controlling mirror M which is on a linear translational stage. ...”

j) Supplementary information, page 3, it is written: "...where the hologram design allowed us to investigate the relationship between any two condensates...";. To which relationship are referring the authors to? To the phase relationship?

The hologram design is a mask which allows the PL from the condensates under investigation to reach the CCD but the PL from the rest of the system is discarded. The PL is observed in Fourier-space which reveals the mutual coherence between the two condensates. The text has been adjusted accordingly to read:

“...the hologram design allowed us to investigate the first-order mutual coherence function between any two condensates...”

k) Supplementary information, page 3, it is written: “...and the extracted real-space magnitude of the polygon using...” What is meant by magnitude of the polygon?

We thank the reviewer for his comments regarding our insufficient text description. With ‘magnitude’ we were referring to the magnitude of the complex-valued order parameter Ψ , which is extracted via homodyne interferometry, i.e. resonant to the reference (seed) laser.

There have been minor changes to this paragraph in page 3 in the supplementary information to make this clearer.

l) There is a work also on polariton condensates about the influence of particle flow on the appearance of topological defects which maybe is relevant for the subject presented here and that is not quoted, namely the work in Optics Express 20, 16366 (2012).

We thank the reviewer for pointing out this work and the reference has been added.

Response to Reviewer 2:

Reviewer #2 (Remarks to the Author):

The paper by Cookson et al examines the vortices which can be realised in lattices of non-resonantly pumped polariton condensates. They find that, in cases where geometric frustration is present, complex vortices with non-trivial winding numbers can be observed and thoroughly characterised. They show how the average behaviour is well described by a simple GPE model and also that the representative physics of single shot measurement can be obtained from the same calculations. The experiments undertaken here are really beautiful and the level of control shown is impressive. The results represent a significant improvement over previous similar experiments showing much finer control over a larger lattice. This is a valuable step towards utilising these types of device for analogue computing applications. With these points in mind I recommend publication in Nature Communications.

There are a few minor points which the authors may want to consider before publication:

The discuss of the model they use is entirely in the supplementary information. It would be useful to at least have the GPE in the main text so that readers can easily see what effects have been included

We thank the reviewer for high praise of our work. We have now moved previous Eqs. (S2) and (S3) from the supplemental, and their description, into the main text as Eqs. (1) and (2).

It would be useful to have slightly more discussion about the discrepancies between the experiment and the theory results. Some of this is really just to do with the stochastic nature of the observations, but some is more systematic, for example the asymmetry and slight disagreement in the experimental results of Figs 3,4,5 (j).

We thank the referee for bringing this interesting question up. Firstly, we point out that the model assumes perfectly symmetric conditions whereas experiment is always subject to some error in the position and intensity of the excitation spots, and sample disorder. Second, we have only included a phenomenological energy relaxation mechanism Λ whereas better agreement between observations and

theory can possibly be achieved by including directly the dynamics of the exciton reservoir into the energy relaxation of the polaritons [M. Wouters et al., Phys. Rev. B **82**, 245315 (2010)].

Given the abovementioned, 100% agreement between the 2DGPE and experiment is difficult to achieve which is manifested in some quantitative differences between numerics and experiment. Nonetheless, our main interpretation on vorticity arising from frustration remains valid and numerics still provide good qualitative agreement. We have added the above discussion into the manuscript on page 15.

List of changes:

1. We have changed the symbol for the vortex winding from “ w ” to “ m ” to remain consistent with our Bloch analysis.
2. We have added Ref.[28] to the introduction and simplified the discussion in the following sentences.
3. We have added Refs.[45, 47-51] to the introduction which concern work on polariton vorticity.
4. According to the suggestion of reviewer 2, we have moved previous supplemental equations (S2) and (S3) into the main text as equations (1) and (2), along with the text explaining our model
5. Removed inset from Fig.3(j) and improved our phase extraction in panels (c) and (d).
6. Removed panel (b) from previous Figs.4 and 5 showing the interferogram from the central region of the polygon and now made new Figs.4 and 5 in a 3x3 panel form.
7. In Fig.5(h) we now only plot the phase difference between condensates 0-1, 1-2, 2-3, and 0-3. We have excluded the analysis on the phase of condensate 4 since it does not have a well defined phase due to destructive interference splitting it. This does not change the message of Fig.5 and only eliminates ambiguity on how to define the phase for “condensate 4”.
8. We have expanded the discussion in the supplemental section S1 (after the first paragraph) on our experimental techniques in response to reviewer 1. We have added Ref.[S6].
9. We have clarified the different acronyms in the caption of Fig.S1.
10. We have added our answer to the second comment of reviewer 2 as a new paragraph on page 15.
11. Few technical/grammatical corrections across the manuscript and in the list of references.

Reviewers' Comments:

Reviewer #1:

Remarks to the Author:

I believe that the authors have satisfactorily answered all the referees' criticisms. However, there are a couple of issues that they should address before publication is granted:

a) In their rebuttal it is mentioned that they have "...added coloured markers on the black circles in Figs. 3(d,h), 4(c,f) and 5(d,h)...", however, there is neither a mention to the change in Fig. 3d, which is not the same as the one in the original version, nor to Fig. 3h, which have the colours reversed with respect to the original one. Could the authors, please, comment on it?

b) In their rebuttal it is explained the origin of the oscillations seen in the phase of the simulations in Fig 4j (new Fig. 4i), I think that it would be convenient to implement in the manuscript this comment on the extra modulation of the phase.

c) I agree on that Eqs. (S2) and S(3) are moved to the main text to Eqs. (1) and (2), but I would like to suggest that the now-labelled "II. 2D GROSS-PITAEVSKII SIMULATION PARAMETERS" section in the Supplementary Information should start with some text as, for example: "We specify here the values used for the parameters appearing in Eqs.(1) and (2) of the main text."

Reviewer #2:

Remarks to the Author:

The authors have addressed all of my previous points and I recommend the manuscript for publication in Nature Communications in its present form

Reviewer #1 (Remarks to the Author):

I believe that the authors have satisfactorily answered all the referees' criticisms. However, there are a couple of issues that they should address before publication is granted:

a) In their rebuttal it is mentioned that they have "...added coloured markers on the black circles in Figs. 3(d,h), 4(c,f) and 5(d,h)...", however, there is neither a mention to the change in Fig. 3d, which is not the same as the one in the original version, nor to Fig. 3h, which have the colours reversed with respect to the original one. Could the authors, please, comment on it?

We apologise for not stating this explicitly earlier. During review of the manuscript the analysis of the phase extraction was rechecked and we felt that minor experimental discrepancies with respect to the theory were reduced through a slight modification to the subtracted reference wave during the phase extraction. The simulation was then matched to the new experimental phase map [see Fig.3(d) and Fig.3(h)] by a simple global phase rotation $\Psi \rightarrow \Psi e^{i\phi}$ which has no effect on the presence of the vortex or the message of the study.

b) In their rebuttal it is explained the origin of the oscillations seen in the phase of the simulations in Fig 4j (new Fig. 4i), I think that it would be convenient to implement in the manuscript this comment on the extra modulation of the phase.

We agree with the reviewer and have added this discussion to the text on page 13.

c) I agree on that Eqs. (S2) and S(3) are moved to the main text to Eqs. (1) and (2), but I would like to suggest that the now-labelled "II. 2D GROSS-PITAEVSKII SIMULATION PARAMETERS" section in the Supplementary Information should start with some text as, for example: "We specify here the values used for the parameters appearing in Eqs.(1) and (2) of the main text."

This has been added to the text.

Reviewer #2 (Remarks to the Author):

The authors have addressed all of my previous points and I recommend the manuscript for publication in Nature Communications in its present form

We thank the reviewer for their expert feedback and are appreciative of the recommendation.

Reviewers' Comments:

Reviewer #1:

Remarks to the Author:

The authors have addressed all my previous points and, in particular, they have clarified the changes made in Fig. 3, therefore I recommend publication in Nature Communications in its present form.